# Design, Synthesis, and Evaluation of Niclosamide Analogs as Therapeutic Agents for Enzalutamide-Resistant Prostate Cancer

**DOI:** 10.3390/ph16050735

**Published:** 2023-05-12

**Authors:** Borui Kang, Madhusoodanan Mottamal, Qiu Zhong, Melyssa Bratton, Changde Zhang, Shanchun Guo, Ahamed Hossain, Peng Ma, Qiang Zhang, Guangdi Wang, Florastina Payton-Stewart

**Affiliations:** 1Department of Chemistry, Xavier University of Louisiana, 1 Drexel Drive, New Orleans, LA 70125, USA; bkang@xula.edu (B.K.); mmottama@xula.edu (M.M.); qzhong@xula.edu (Q.Z.); czhang1@xula.edu (C.Z.); sguo@xula.edu (S.G.); qzhang1@xula.edu (Q.Z.); 2RCMI Cancer Research Center, Xavier University of Louisiana, 1 Drexel Drive, New Orleans, LA 70125, USA; mbratton@xula.edu (M.B.); hahamed@xula.edu (A.H.); pma@xula.edu (P.M.)

**Keywords:** niclosamide analogs, enzalutamide-resistant prostate cancer, androgen receptor variant 7 (AR-V7) down-regulators, structure–activity relationship (SAR)

## Abstract

Niclosamide effectively downregulates androgen receptor variants (AR-Vs) for treating enzalutamide and abiraterone-resistant prostate cancer. However, the poor pharmaceutical properties of niclosamide due to its solubility and metabolic instability have limited its clinical utility as a systemic treatment for cancer. A novel series of niclosamide analogs was prepared to systematically explore the structure–activity relationship and identify active AR-Vs inhibitors with improved pharmaceutical properties based on the backbone chemical structure of niclosamide. Compounds were characterized using ^1^H NMR, ^13^C NMR, MS, and elemental analysis. The synthesized compounds were evaluated for antiproliferative activity and downregulation of AR and AR-V7 in two enzalutamide-resistant cell lines, LNCaP95 and 22RV1. Several of the niclosamide analogs exhibited equivalent or improved anti-proliferation effects in LNCaP95 and 22RV1 cell lines (**B9**, IC_50_ LNCaP95 and 22RV1 = 0.130 and 0.0997 μM, respectively), potent AR-V7 down-regulating activity, and improved metabolic stability. In addition, both a traditional structure–activity relationship (SAR) and 3D-QSAR analysis were performed to guide further structural optimization. The presence of two -CF_3_ groups of the most active **B9** in the sterically favorable field and the presence of the -CN group of the least active **B7** in the sterically unfavorable field seem to make **B9** more potent than **B7** in the antiproliferative activity.

## 1. Introduction

Prostate cancer (PC) is the most frequently diagnosed cancer in males with nearly 1.3 million new cases diagnosed in 2018, and the second leading cause of cancer death worldwide [1]. Its development and progression rely on the androgen receptor (AR), which is expressed in the majority of androgen-independent or hormone-refractory PC. The inhibition of AR activity through modulation of AR-dependent signaling pathways has become the backbone endocrine therapy to arrest PC progression [2,3,4,5]. However, almost all treated patients develop resistance to androgen deprivation, resulting in the development of castration-resistant prostate cancer (CRPC). Recent studies have shown that further hormonal manipulation can result in impressive disease control even after the progression of ADT, and thus, many patients with CRPC would respond to further hormonal manipulation [6]. This was demonstrated in phase III randomized trials showing improved survival for patients receiving the CYP17 hydroxylase inhibitor abiraterone and the second-generation AR antagonist enzalutamide [7,8,9,10]. These drugs proved that AR remains a clinically relevant therapeutic target in CRPC. Unfortunately, the clinical benefits of abiraterone and enzalutamide therapies are limited by the inevitable development of resistance to both drugs [11,12]. Numerous mechanisms have been demonstrated to drive disease progression upon AR-targeted therapy [13], of which the truncated and constitutively active AR variants (AR-Vs) due to alternative mRNA splicing have emerged as an important and common driver of drug resistance as they are frequently upregulated in castration-resistant vs. hormone-responsive tumor tissues [14,15,16,17,18]. In AR-Vs, the ligand binding domain (LBD) is truncated, and capable of ligand-independent activation of androgen-responsive element (ARE)-driven reporters in the absence of androgen. While AR-Vs confer ligand independence, the downstream signaling remains critical in the proliferation of CRPC. It offers a clinical opportunity for novel agents targeting AR-Vs expression to prohibit CRPC tumor growth.

As an antihelminthic drug since the 1960s, niclosamide is known for its safety, which makes it an attractive template for drug development. Niclosamide has demonstrated notable cytotoxicity against tumor cell lines and cancer stem cells (CSCs) with minimum effect on normal cells [19]. Furthermore, a broad inhibitory activity towards tumor-involving protein signaling pathways, such as signal transducer and activator of transcription 3 (STAT3), Notch, nuclear factor κB (NF-κB), and Wnt/β-catenin, are reported in the past decade. In addition, niclosamide has been identified as a downregulator of AR-V7 protein, while reducing its recruitment to promoter regions of target genes and exhibits robust efficacy in enzalutamide [20] and abiraterone [21] resistant prostate cancer models.

Despite its promising biological activities, however, niclosamide has poor pharmacokinetics (PK) properties, presenting obstacles in the pathway to clinical usage as a systemic anticancer agent. The inappropriate PK properties are primarily due to its poor solubility, rapid metabolism, and absorption profile [22,23,24]. Previous studies in mice revealed that the drug was rapidly cleared from systemic circulation and afforded only modest steady-state plasma concentration and tumor tissue distribution after a repeated daily oral dose of 200 mg/kg [25,26]. It could be attributed to the metabolic instability of the substituents in the molecule, such as the *ortho* phenolic hydroxyl (*o*-OH) and *para* nitro (*p*-NO_2_) group. However, structure–activity relationship analysis indicated both substituents are essential for biological activities [26]. Thus, it is of interest to determine if the replacement of these groups with more inert and physiologically less labile ones might improve bioavailability while maintaining sufficient potency. So far, efforts including structural modifications [25,26,27,28,29] and novel drug delivery strategies [30,31] have not yet resulted in significantly improved compounds with greater activities and better pharmaceutical properties than niclosamide.

This study was undertaken to systematically explore the structure–activity relationship of niclosamide analogs to develop a novel series of potent nonsteroidal AR-Vs inhibitors with improved pharmaceutical properties based on the backbone chemical structure of niclosamide. Various substituents, with a wide range of electron-donating/withdrawing properties, size, and physiological stability were tested for their effects on the inhibitory activities of the resulting niclosamide analogs against two AR-V7 expressing prostate cancer cell lines, 22RV1 and LNCaP95. The 22RV1 and LNCaP95 cell lines are well characterized as stably expressing high-level AR and AR-V7 and are resistant to enzalutamide [31]. Selected active analogs were also tested for their ability to downregulate the AR-V7 protein for mechanistic confirmation. The structure–activity relationship analysis provided useful guidance on the optimization of niclosamide analogs as a potential treatment for enzalutamide-resistant prostate cancer.

## 2. Results and Discussion

### 2.1. Chemistry


*Design*


Several amide and imide-containing small molecules represented by niclosamide and enzalutamide, respectively, have been developed as AR competitive inhibitors (Figure 1). Their common molecular feature is an aniline fragment with electron-withdrawing groups (EWGs). They compete with androgen for the binding of AR, consequently blocking the action of androgens of adrenal and testicular origin that stimulate the growth of normal and malignant prostatic tissues. This blockade may result in growth arrest or transient tumor regression through inhibition of androgen-dependent DNA and protein synthesis. However, few in-depth studies of the structure–activity relationship (SAR) of these compounds have been reported. Previous studies indicate that the *p*-NO_2_ group of niclosamide is crucial for biological activity without a detailed explanation [26]. Therefore, our primary goal is to determine how and the extent to which the nitro group may affect the activity. Considering the potential cytotoxicity of nitro compounds, our secondary approach is to find a preferable alternative group and further explore the contribution of substituents on both phenyl rings to evaluate the inhibitory effect. Compound **A** (Figure 2) was designed to evaluate the effect of other substituents on the activities by retaining the *p*-NO_2_. Meanwhile, the design of compound **B** (Figure 2) aimed to evaluate the significance of the *p*-NO_2_ group by removing it or replacing it with other EWGs.

Secondly, a pseudo-six-member ring might be formed via an intramolecular hydrogen bond (HB) between the phenolic hydroxyl group and the amide fragment. Under this circumstance, the relative position and angle of the two aromatic rings would be fixed. The AR inhibitory effect of niclosamide may benefit from this specific configuration. Thus, the design of compound **C** (Figure 2) focused on determining how and the extent to which the pseudo-six-member ring conformation affects the AR inhibition by removing *o*-OH or replacing it with other groups.


*Synthesis*


The general synthetic route of compounds **A1**–**A20**, **B1**, **B3**–**B16**, and **C1**–**C5** is illustrated in Figure 1. The chlorination of commercially available substituted benzoic acid **S** was carried out in thionyl chloride and anhydrous THF with high yields. The acid chloride **M** reacted with substituted aniline in the presence of a catalytic amount of DMAP to obtain the desired compounds with yields arranged from 55% to 95%. Compounds **B2** and **C6**–**C8** were obtained by the reduction reactions as depicted in Figure 2 and Figure 3, respectively.

### 2.2. Biological Screening


*Antiproliferative activities in enzalutamide resistant prostate cancer cells.*


The synthesized compounds were first screened in 22Rv1, an enzalutamide-resistant cell line at concentrations of 10 and 1.0 μM, respectively. Following this, LNCaP95 and 22RV1 cells were treated with selected active compounds at various doses ranging from 1 nM to 10 μM to determine their IC_50_ values. Results are summarized in Table 1.

The majority of compounds exhibited effective anti-proliferative activities in the enzalutamide-resistant cell lines, LNCaP95 and 22RV1. In all, 7 compounds (**A6**, **A20**, **B1**, **B6**, **B9**, **B16**, and **C8**) showed equivalent or improved anti-proliferation potency compared with that of niclosamide (Table 1). Notably, the best two analogs, **B9** and **B16**, lowered the IC_50_ value from 0.372 μM to 0.130 μM and 0.113 μM in LNcap95, from 0.284 μM to 0.0997 μM and 0.111 μM in 22RV1, respectively. The structures and cell-based screening data of all tested compounds are listed in the Appendix A. 


*Downregulation of AR-FL and AR-V7 in LNCaP95 and 22RV1 cells*


Downregulation of AR-V7 expression in enzalutamide-resistant prostate cancer cells is an important mechanistic measure of the anti-proliferative activities of niclosamide. We next determined the ability of the most potent niclosamide analogs to downregulate the expression of AR-V7 and AR-full length (AR-FL) in LNCaP95 and 22RV1 cells expressing AR-V7 splice variants. Niclosamide induced degradation of both AR-FL and AR-V7 in LNCaP95 cells (Figure 3) with comparable potency (IC_50_ = 0.633 and 0.644 μM, respectively, Table 2). In comparison, active niclosamide analogs **B1**, **B3**, **B9**, **and B14** all showed strong activity in downregulating the AR-FL and AR-V7 in LNCaP95 cells (Figure 3, Figure 4 and Figure 5). In 22RV1 cells, niclosamide and analogs showed differentiated activity in the downregulation of AR-FL and AR-V7. While niclosamide was ten-fold more active in AR-FL (IC_50_ = 0.707 μM) than in AR-V7 downregulation (IC_50_ = 6.98 μM), **B3** showed greater activity in AR-V7 than in AR-FL downregulation (9.93 vs 0.354 μM). Similar to niclosamide, however, **B1**, **B9**, and **B14** all exhibited greater potency in degrading AR-FL than AR-V7.

### 2.3. Structure–Activity Relationship (SAR) Analysis


*The effect of substituents R^1^*


We first examined the effect of the halogen on ring A. All **A** compounds, except for **A11** and **A19**, exhibit potent antiproliferative activity in LNCaP95 and 22RV1 prostate cancer cell lines. The 5-Cl analog (**A10**–**A18**) showed slightly greater activity than their 4-Cl counterparts (**A1**–**A8**). On the other hand, replacing chlorine with bromine resulted in a dramatic loss of activity (**A19**). This exception drew our attention and would be separately discussed.

We next examined whether the combination of *o*-Cl and *p*-NO_2_ on ring B affects the contribution of 5-substitution on ring A. Several electron-withdrawing groups (EWGs) and hydrogen bond receptors were introduced on ring B. At the same time, the 5-Cl on ring A was replaced by bromine, fluorine, nitrile, and trifluoromethyl, aimed at finding preferable substitutes as alternatives to chlorine.

Compared with the compounds with 5-Cl, the modification on ring A maintained the antiproliferative activity. Meanwhile, their activity seems slightly improved with the enhancement of the electron-withdrawing capacity of the substitutions. The average IC_50_ value of modified compounds was observed at around 300 nM, compared with 500–600 nM for 5-Cl compounds. Specifically, the **B9** (5-F) and **B16** (5-CF_3_) compounds exhibited impressive potency with IC_50_ of around 100 nM in both LNCaP95 and 22RV1 cell lines. With a few exceptions, the majority of **A** and **B** compounds retained the antiproliferative activity when modifications made on ring A were tolerated to various degrees.

In summary, we observed that the R^1^ contribution to the anti-proliferation potency of the analogs in two enzalutamide-resistant cell lines, and 5-F and 5-CF_3_ on ring A are the preferable substitutes for optimal activity.


*The effect of substituents R^2^*


The first modification of R^2^ included replacing the *o*-Cl with other EWGs (**A1**, **A10**, **A11**), electron-donating groups (EDGs) (**A2**, **A3**, **A12**, **A13**, **A20),** and transferring the groups from *ortho-* to *para-* position (**A4**–**A8**, **A14**–**A18**). The IC_50_ values of the majority of **A** compounds were measured around 800 nM without significant improvement. However, both the modifications of reducing *p*-NO_2_ to *p*-NH_2_ (**B2**) and replacing *p*-NO_2_ with *p*-Cl resulted in a complete loss of activity. This confirmed that *p*-NO_2_ was crucial and contributed more to the antiproliferative activity of the analogs than other substitutes on ring B.

The second modification takes into consideration the possible cytotoxicity and poor solubility of nitro compounds and seeks to identify an alternative to *p*-NO_2_. Initially, we attempted to replace the nitro group with nitrile (**B3**) and trifluoromethyl (**B1** and **B4**). Herein, *p*-CF_3_ compounds exhibited improved anti-proliferation with an IC_50_ value of around 200 nM in two tested cell lines. After that, more -CF_3_-containing compounds were synthesized (**B5**–**B11**), and two compounds with *p*-CN were obtained as comparisons (**B12** and **B13**). As expected, the introduction of –CF_3_ slightly improved the activity of the compound. Of all **B** compounds, **B9** (3′, 4′-diCF_3_) showed the greatest potency (IC_50_ in LNCapN95 and 22RV1 = 0.130 and 0.0997 μM, respectively) followed by **B6** (2′-F, 4′-CF_3_; IC_50_ in LNCapN95 and 22RV1 = 0.185 and 0.196 μM, respectively) and **B13** (2′-F, 4′-CN; IC_50_ in LNCapN95 and 22RV1 = 0.231 and 0.184 μM, respectively). Based on these data, *p*-CF_3_ emerged as the preferable alternative to *p*-NO_2_, and 3′, 4′–diCF_3_ is the best combination on ring B. Notably, compared with their nitro counterparts, the trifluoromethyl compounds showed limited tolerance of other groups on ring B.

Once we determined the suitable groups on ring B, we synthesized three additional compounds, which replaced the 5-Cl on ring A (**B1**) with 4, 5-diF (**B14**), 5-CN (**B15**) and 5-CF_3_ (**B16**), respectively. It seems that the stronger EWGs on ring A, the better activity of the overall molecule.


*The effect of o-OH on ring A*


Niclosamide could form a pseudo-six-member ring via an intramolecular hydrogen bond (Figure 2). In the specific configuration, *o*-OH is essential for bridging ring A and the amide fragment. For evaluating the contribution of the hydroxyl group on activity, **C** compounds without *o*-OH on ring A were synthesized and tested.

The anti-proliferation activities of the compounds with a trifluoromethyl group on ring B (**C1**, **C3,** and **C5**) were retained. In contrast, removing the hydroxyl group of the nitro analogs caused a significant loss of activity (**C2** and **C4**). Furthermore, when the hydroxyl group is transformed to triflate (**C6** and **C8**) and acetate (**C7**) to block the intramolecular pseudo ring formation, the activities of triflate derivatives were slightly improved to different degrees (**C6** vs. **A7**, **C8** vs. niclosamide) and the acetate analog showed comparable activity (**C7** vs. niclosamide). This observation suggests that *o*-OH played an equally essential role as *p*-NO_2_ on ring B in conferring activities as a hydrogen bond donor in the pseudo-six-member ring configuration. However, in the trifluoromethyl compounds, the role of *o*-OH is reduced to that of an EWG only.

Previously, we also observed two exceptions in structure–activity relationships (**A11** and **A19**). The successful modifications in –CF_3_ derivatives did not result in the desired activity increase in these two nitro compounds. Based on observed structure–activity relationships, we speculated that the *p*-NO_2_ compounds represented by niclosamide have a different mechanism of action from the *p*-CF_3_ compounds represented by **B9** and **B16**, although they exhibited equivalent anti-proliferation activities in the two enzalutamide-resistant cell lines. Generally, the structure–activity relationship analysis results can be illustrated in Figure 6.

### 2.4. 3D-QSAR Analysis

3D-QSAR methodology yields indirect information about the ligand binding characteristics to a receptor molecule from the correlation between the biological activity of a set of compounds and their 3D structures. The model included steric, electrostatic, hydrophobic, hydrogen bond acceptor, and hydrogen bond donor filed descriptors. The observed and predicted pIC_50_ by the Gaussian-based QSAR calculations and the residuals for the training and test sets are presented in Table 3, and the correlations between the observed and predicted pIC_50_ are depicted in Figure 7. The r^2^, q^2,^ and the optimum number of components are 0.76, 0.75, and 5, respectively. Eight compounds were randomly selected to test the predicting capability and they were not included in the process of the construction of the QSAR model. The predicted IC_50_ values of the test sets were consistent with the experimental IC_50_ in a statistically tolerable error range.

The best model with an optimal partial least square (PLS) factor of 5 yielded a squared Pearson correlation coefficient of R^2^ = 0.76 for the training set, a Pearson correlation coefficient of Pearson-R = 0.87 for the test set, and a predictive squared correlation coefficient of Q^2^ = 0.75. To determine the stability of the model, the scrambled correlation coefficient was obtained using scrambled biological activities, and the R^2^_Scrambled_ was 0.55. The model resulted in acceptable correlation coefficients.

Fractions of the Gaussian field contributions for the PLS factor 5 are steric 33%, electrostatic 13%, hydrophobic 18%, hydrogen bond acceptor 29%, and hydrogen bond donor 7% fields. The steric and h-bond acceptor fields were identified as the major constituents of the antiproliferative activity of the compounds against 22RV1 cells, followed by the hydrophobic, and electrostatic fields. The contour maps of the most active (**B9**) and the least active (**B7**) compounds are shown in Figure 8. The contour maps provide information about the nature and spatial arrangement of the pharmacophoric fields of niclosamide derivatives, which are complementary to the binding site of the target protein. The steric fields are shown in Figure 8a, where the green contours denote areas where bulky groups favor biological activity, whereas the yellow contours represent the areas where bulky groups reduce activity. In compound **B9,** the steric field at positions 3′ and 4′ of phenyl ring B is occupied by bulky -CF_3_ groups, whereas the same positions in compound **B7** are occupied by H and -CF_3_, which makes **B9** relatively more active than **B7**. Similarly, in the sterically less favored position, the presence of -CN at position 2 of ring B may also make **B7** less active than **B9,** which has an H at position 2 of ring B. The effect of the electrostatic field is shown in Figure 8b. The blue color contours indicate that electropositive substituents at these regions would increase the activity, while red contours indicate the regions where electron-rich substituents are beneficial for the activity. The hydrophobic fields (Figure 8c) are represented by yellow (favored) and white (disfavored) contours. Similarly, the hydrogen bond donor fields (Figure 8d) are indicated by blue-violet (favored) and cyan (unfavored) contours, and the hydrogen bond acceptor fields (Figure 8e) are represented by red (favored) and magenta (unfavored) contours. The presence of the nitrile group of **B7** toward the unfavorable h-bond acceptor region may also lower the activity of **B7**.

Based on these results, the contour maps offer valuable information necessary to understand the relationship between the physicochemical structure and antiproliferative activity. Because the anti-proliferative activity of the most active niclosamide analog (**B9**) is still in the micromolar range, the chemical and biological information provided is valuable and can be exploited to design novel, more potent agents against enzalutamide-resistant prostate cancer cells.

In addition, we analyzed in silico physicochemical descriptors and pharmaceutically significant properties of all the 40 compounds (see Appendix A), which determine their absorption, distribution, metabolism, and excretion (ADME) properties. The predicted pharmaco-kinetic properties are within the acceptable range desired for human chemotherapeutic uses. Thus, the ADME analysis reveals that these compounds have significant potential as drug-like molecules to be used for chemotherapeutic treatments.

## 3. Materials and Methods

### 3.1. General Information

All reagents, solvents, and analytical standards were purchased from Sigma-Aldrich (St. Louis, MO, USA), Fisher Scientific (Fairfield, NJ, USA), AK Scientific (Union City, CA, USA), and CombiPhos Catalysts (Princeton, NJ, USA) and used without further purification, unless otherwise specified. The crude products or reaction mixtures were purified on a Combi *Flash* Rf flash chromatography system (Teledyne Isco, NE, USA) with a silica gel flash column (40–60 μm, Bonna-Agela Technologies, DE). The yield was recorded as an isolated yield. The purity of all biologically evaluated compounds is greater than 95%. ^1^H and ^13^C NMR spectra were recorded on an Agilent 400-MR NMR spectrometer (400 MHz and 100 MHz, respectively). The data were processed using MestReNova NMR software (School of Chemistry, University of Bristol, Bristol, UK). Chemical shifts are reported as parts per million (ppm) relative to TMS (0.00 ppm) or residual undeuterated solvent signal. HRMS spectra data were collected on a Thermo LTQ Orbitrap-XL mass spectrometer in positive ion mode. The tested compound was confirmed to be >95% pure by HPLC.

### 3.2. General Procedures for the Preparation of **A1**–**A20**, **B1**–**B16**, and **C1**–**C5**

A solution of SOCl_2_ in anhydrous THF (5 mL dissolved in 5 mL anhydrous THF) was added dropwise at 0 °C under N_2_ to the solution of substituted benzoic acid (1.00 mmol) in anhydrous THF (15 mL). The reaction mixture was stirred at r.t. for 1.5 h. The solvent was removed by reduced pressure distillation to obtain white to off-white solid or semi-solid. The residue was dissolved in anhydrous THF (10 mL) and added to the mixture of substituted aniline (1.1 eq.) and DMAP (cat.) in anhydrous THF dropwise at 0 °C under N_2_. The reaction mixture was stirred at r.t. until the benzoic acid chloride was fully consumed. The solvent was removed under vacuum and the residue was suspended in EA (20 mL), washed with HCl (aq., 2 M, 10 mL × 3), sat. NaHCO_3_ (aq., 10 mL × 2), and sat. NaCl (aq., 10 mL), and dried over NaSO_4_. The mixture was concentrated and purified by flash column chromatography (silica gel, 60 Å) with a gradient eluent of hexane and EA.

### 3.3. Procedures for the Preparation of **B2**

The reaction mixture of niclosamide (1.0 g, 3.05 mmol), reduced iron powder (0.85 g, 15.25 mmol), and HCl (aq., 1 mL) in methanol (8 mL) and ethanol (20 mL) was refluxed for 3.0 h. The suspension was filtered. The filter cake was washed with methanol (×3). The combined filtrates were evaporated and purified by flash column chromatography on silica gel with a gradient eluent of hexane and EA to obtain the desired compound as a pale-yellow solid (0.86 g, 95.1%).

### 3.4. Procedures for the Preparation of **C6**–**C8**

To the solution of niclosamide (1.00 mmol) and pyridine (1.00 mmol) in DCM was added acetic anhydride or trifluoromethanesulfonic anhydride (1.10 mmol) dropwise at 0 °C. The reaction mixture was stirred for 1.0–1.5 h. The reaction mixture was diluted with DCM (diluted approximately to 25 mL), washed with HCl (aq., 2 M), saturated Na_2_CO_3_ (aq.) and NaCl (aq.), and dried over MgSO_4_. The organic fraction was concentrated and purified by flash column chromatography on silica gel with a gradient eluent of hexane and EA to obtain the desired compound as an off-white solid.

### 3.5. Antiproliferative Assays

22Rv1 cells were obtained from ATCC. LNCaP95 cells were provided by Dr. Yan Dong’s laboratory (Tulane University School of Medicine). LNCaP95 cells were maintained in an RPMI1640 medium with charcoal-stripped serum (CSS). 22Rv1 cells were maintained in an RPMI1640 medium with 10% FBS. Cells were seeded in 96-well plates (5 × 10^3^ cells/well) containing a 50 μL growth medium for 24 h. After medium removal, a 100 μL fresh medium containing individual compounds at different concentrations was added to each well and incubated at 37 °C for 72 h. Then, 20 μL of resazurin was added for 2 h before recording fluorescence at 560 nm (excitation) and 590 nm (emission) using a Victor microtiter plate fluorometer (Perkin-Elmer, Waltham, MA, USA). The IC_50_ was defined as the compound concentration required for inhibiting cell proliferation by 50%, in comparison with cells treated with the maximum amount of DMSO (0.25%) and considered as 100% viability.

### 3.6. Western Blot Analysis

Total AR and AR-V7 protein levels were determined using automated Western blotting (Wes Simple Western Analysis, ProteinSimple, San Jose, CA, USA). WES analyses were performed following the WES user manual. Briefly, LNCaP95 and 22RV1 cells were plated in a 12-well plate at a density of 2.5 × 10^5^ cells per well. Cells were treated with either DMSO, niclosamide, or test compounds at the indicated concentrations for 24 h, and protein extraction from cells was conducted using MPER lysis buffer (Pierce) containing protease and phosphatase inhibitors (Thermo Scientific, Waltham, MA, USA). Samples were mixed with a master mix (ProteinSimple, San Jose, CA, USA) to give a final concentration of 0.2 mg/mL total protein, 1 × sample buffer, 1 × fluorescent molecular weight markers, and 40 mM DTT. Samples were heated at 95 °C for 5 min followed by centrifugation. Samples, blocking solution, primary antibodies, horseradish peroxidase-conjugated secondary antibodies, chemiluminescent substrate, and separation and stacking matrices were loaded into designated wells in a 384 well plate. After plate loading, fully automated electrophoresis and immunodetection took place in the capillary system. Proteins were separated by molecular weight at 375 V for 25 min, and primary and secondary antibodies were incubated for 30 min. The AR (Santa Cruz, Santa Cruz, CA, USA), AR-V7 (Santa Cruz, Santa Cruz, CA, USA), and actin (Novus, cat #NB600-503) antibodies were diluted in a proprietary antibody diluent at a 1:50 dilution ratio. Chemiluminescence was captured by a charge-coupled device camera, and the digital image was analyzed using Compass software 6.2.0 (ProteinSimple). The amount of each protein, relative to total protein content, was calculated based on peak areas. Total AR and AR-V7 levels were normalized to actin.

### 3.7. 3D-QSAR Study

3D-QSAR relates the variation of biological response in a series of related compounds to the variation in their 3D chemical structures. In the drug design and discovery area, this method has been widely used for the design of novel more potent small-molecule drug candidates. Anti-proliferative activity of 40 niclosamide analogs against 22RV1 cell lines was considered for building the QSAR model. All the compounds were prepared using the Schrödinger Suite followed by energy minimization with the OPLS4 force field [32,33]. Superposition of the molecules was carried out by aligning the common benzanilide present in the molecules on the most active compound. A total of 80% of the data set was randomly selected as the training set and the remaining 20% as the test set. For the PLS regression analysis, the pIC_50_ values (−log IC_50_) were considered as the dependent variables, and the Gaussian field descriptors as the independent variables in 3D-QSAR models. We used the field-based QSAR tool of the Schrodinger Suite to develop Gaussian-based QSAR models [34]. The model was built by constructing a cubic lattice with 1 Å grid spacing and extended by 3 Å beyond the training set limits. The electrostatic and steric force fields were truncated at ±30 kcal/mol, and variables with a standard deviation <0.01 were eliminated. Subsequently, the Gaussian-based steric, electrostatic, hydrophobic, hydrogen bond acceptor, and hydrogen bond donor fields were calculated to examine their contributions to the activity. To determine the stability of the model, progressive scrambling on the model was performed. Finally, in silico ADME properties of all the niclosamide analogs were obtained by using QikProp (Schrodinger Suite) [35].

## 4. Conclusions

A new series of niclosamide analogs has been rationally designed and synthesized to evaluate their activities against enzalutamide-resistant prostate cancer cells. Compared with the prototype compound niclosamide, the new analogs that exhibited equivalent or improved antiproliferative on 22RV1 and LNaP95 cell lines (**B9**, IC_50_ value of 0.130 and 0.0997 μM in LNCaP95 and 22RV1 respectively) were found to potently downregulate AR-V7 and AR-FL protein expression. A traditional structure–activity relationship (SAR) and 3D-QSAR analysis were performed to guide further structural optimization. We found that a preferable alternative substituent (trifluoromethyl) to the nitro group in niclosamide conferred improved anticancer efficacy and broader tolerance to structural modifications. The contribution of varying substituents to the overall antiproliferative activity of the niclosamide in enzalutamide-resistant cell lines has been determined. Further, the 3D-QSAR studies offered valuable information about different types of fields which can enhance the activity of niclosamide derivatives as anti-PC agents. The steric field followed by the hydrogen bond acceptor field seems to be the major contributing descriptor fields in exhibiting the antiproliferative activity of niclosamide analogs in 22RV1 cell lines.

## Data Availability

Data is contained within the article and Appendix A.

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
