# Peer review of "Design, Synthesis, and Evaluation of Niclosamide Analogs as Therapeutic Agents for Enzalutamide-Resistant Prostate Cancer"

_pharmaceuticals, 2023, doi:10.3390/ph16050735_

Round 1

Reviewer 1 Report

The manuscipt entitled "Design, Synthesis, and Evaluation of Niclosamide Analogs as 2 Therapeutic Agents for Enzalutamide-Resistant Prostate Cancer" describes the synthesis of niclosamide analogues.

The development of novel compounds for the treatment of cancer is an important area of research, and the use of niclosamide analogs to target androgen receptor variants (AR-Vs) is a promising approach. This paper highlights the findings of a study that aimed to identify active AR-V inhibitors with improved pharmaceutical properties based on the backbone chemical structure of niclosamide. The study showed that several of the synthesized compounds exhibited equivalent or improved anti-proliferation effects in enzalutamide resistant cell lines and potent AR-V7 down-regulating activity, indicating their potential as effective treatments for prostate cancer.

 The use of traditional structure-activity relationship (SAR) and 3D-QSAR analysis to guide further structural optimization is also a noteworthy aspect of the study. These analyses provide valuable insights into the relationship between the chemical structure of the compounds and their biological activity, allowing for the development of more effective and targeted treatments.

 Overall, the findings of this study have significant implications for the development of novel therapies for prostate cancer, and the identification of niclosamide analogs with improved pharmaceutical properties is a promising step towards improving the clinical utility of these compounds. 

Some typo errors are found in the manuscript. line no 204

Author Response

Comments 1. Some typo errors are found in the manuscript. line no 204

Response: The following typos are corrected. A comma is added after Thus (line 82), extra spaces removed (111), removed On (line 114), theinhibitory is changed to the inhibitory (line 116), changed compound to compounds (line 133), (b)Substituted is separated with a space (line 143), Compound B is changed to Compound B2 (line 146), C6-C8 is changed to C6-C8 (line 151), mean is changed to the means (line 169), added a period after experiments (line 169), in Table 2 IC50 is changed to IC50 (line 196), changed degree to degrees (line 218), changed reducing of to reducing (line 228), could is changed to can (line 272), physciochemical is changed to physicochemical (line 319), A, B and C compounds were changed to A, B and C (lines 360, 372, and 379), ml is changed to mL (line 362 and 374), 0°C is changed to 0 °C (lines 142, 143, 153, 363, 366, and 381), pyriding is changed to pyridine (line 380), analog is changed analogs (line 444).

Reviewer 2 Report

The present research manuscript entitled “Design, synthesis, and evaluation of niclosamide analogs as therapeutic agents for enzalutamide-resistant prostate cancer” Kang et al. is novel and well-designed. The researchers conducted a number of in vitro studies to prove their claim. The results are interesting, the flow of the manuscript is very good, and the diagrammatic representation of the data is excellent. Overall, the quality of the manuscript is excellent. However, some corrections are needed to improve the overall quality of the manuscript. My comments are as follows.

Comment 1. What are the poor pharmaceutical properties of niclosamide that limit its clinical utility? The authors should discuss this briefly in the abstract as well as in the introduction.

Comment 2. A brief concluding remark should be included in the abstract section.

Comment 3. What about the statistical analysis of data? The authors should provide the statistical statement and the obtained data should be analyzed. Further, almost all the obtained data is mentioned without standard deviation. Each study should be conducted at least three times and the data should be mentioned with standard deviation in the manuscript.

Comment 4.  Antiproliferative assays: The authors should elaborate on this method. This section is confusing. The authors should mention the concentration range. Further, it would be better to conduct this study at 48 h to improve its overall impact.

Comment 5. Overall, the quality of the language is poor and lacks scientific value in many sentences. The authors should revise the manuscript from top to bottom to improve the overall quality of the manuscript. 

Overall, the quality of the language is poor and lacks scientific value in many sentences. The authors should revise the manuscript from top to bottom to improve the overall quality of the manuscript. 

Author Response

Comment 1. What are the poor pharmaceutical properties of niclosamide that limit its clinical utility? The authors should discuss this briefly in the abstract as well as in the introduction.

Response: As mentioned in the Introduction (lines 75-76), the inappropriate PK properties of niclosamide are due primarily to its poor solubility, rapid metabolism, and absorption profile [22-24]. It is now also briefly described in the Abstract.

Comment 2. A brief concluding remark should be included in the abstract section.

Response: A brief concluding remark is now included in the abstract.

Comment 3. What about the statistical analysis of data? The authors should provide the statistical statement and the obtained data should be analyzed. Further, almost all the obtained data is mentioned without standard deviation. Each study should be conducted at least three times and the data should be mentioned with standard deviation in the manuscript.

Response: Statement of “The data given are mean of three independent experiments” are provided for each table of experimental results. Standard deviation is now included in all tabled results.

Comment 4. Antiproliferative assays: The authors should elaborate on this method. This section is confusing. The authors should mention the concentration range. Further, it would be better to conduct this study at 48 h to improve its overall impact.

Response: The synthesized compounds were first screened in 22Rv1, an enzalutamide resistant cell line at concentrations of 10 uM and 1.0 uM, respectively. Following this, LNCaP95 and 22RV1 cells were treated with selected active compounds at various doses ranging from 1 nM to 10 uM to determine their IC50 values. Results are summarized in Tables 1. Treatment lasted 72 hours to ensure optimal impact of cell cytotoxicity of the niclosamide analogs.

Comment 5. Overall, the quality of the language is poor and lacks scientific value in many sentences. The authors should revise the manuscript from top to bottom to improve the overall quality of the manuscript.

Response: We have improved the quality of language throughout the manuscript.

Reviewer 3 Report

The authors described the design, synthesis, and evaluation of niclosamide analogs as therapeutic agents for enzalutamide-resistant prostate cancer. I consider that the manuscript meets all requirements to be published in “Pharmaceuticals” after minor revision. Additional suggestions and/or comments are included:  

(1) See the abstract, lines 22 and 23. I consider that a short paragraph of the most relevant docking data should be included.

(2) See Tables 1 and 2. The standard deviation for all IC50 values should be included.

(3) See Figures 3 and 4. The resolution should be improved.

(4) See 3. D-QSAR. ¿Is it possible to complement these results with ADMET studies?

(5) See the conclusions. The most relevant experimental and theoretical data should be included.  

(6) See Supporting Information. NMR and HRMS spectra of all synthesized compounds should be included.

Author Response

The authors described the design, synthesis, and evaluation of niclosamide analogs as therapeutic agents for enzalutamide-resistant prostate cancer. I consider that the manuscript meets all requirements to be published in “Pharmaceuticals” after minor revision. Additional suggestions and/or comments are included:

(1) See the abstract, lines 22 and 23. I consider that a short paragraph of the most relevant docking data should be included.

Response: A sentence highlighting the most relevant data from the 3D-QSAR studies is included towards the end of the Abstract. (lines 26-28)

(2) See Tables 1 and 2. The standard deviation for all IC values should be included.

Response: Standard deviation data for all IC50 values are now included in Tables 1 and 2.

(3) See Figures 3 and 4. The resolution should be improved.

Response: Improved to the highest possible.

(4) See 3. D-QSAR. ¿Is it possible to complement these results with ADMET studies?

Response: Figures 7 and 8 are the original images in high resolution. We conducted in silico ADMET studies and the results are now included in the supporting information as well as in the Results and Discussion section towards the end of 3D-QSAR studies (lines 324-329). A new reference (35) has been cited and added to the References section. 

(5) See the conclusions. The most relevant experimental and theoretical data should be included.

Response: The most relevant theoretical data is included towards the end of conclusions. (lines 456-458)

(6) See Supporting Information. NMR and HRMS spectra of all synthesized compounds should be included.

Response: For each synthesized compound, 1H NMR, 13C NMR, and HRMS data are provided in the Supporting information. In addition, selected 1H and 13C NMR spectra are provided for 10 synthesized analogs.

Reviewer 4 Report

The authors describe in this manuscript the design and synthesis of a series of niclosamide analogs, which have been evaluated for antiproliferative activity, downregulation of AR and AR-V7 in two enzalutamide resistant cell lines, LNCaP95 and 22RV1. Several compounds were prepared with varieties of substitutions at different positions. Also, careful structure-activity relationship studies and 3D-QSAR analysis have been performed, which led to a most promising compound with good bioactivity. Thus, this reviewer recommends its publication on Pharmaceuticals after minor revisions.   

1. There are many typos and format errors in the manuscript, some of them have been marked in the attached file.

2. Scheme 2, “synthesis of compound B”, however, the scheme is written as synthesis of B2’.

3. Move Figure 6 to the suitable place, such as, line 282.

4. fluorine is an important element for drug discovery. The promising compound in this wrok also containing trifluoromethyl group. Thus, recent reviews on fluorine-containing drugs are recommended to be cited.

5. 1H NMR spectra, there are impurity peaks from 1-2 ppm, clean spectra should be provided.

Have been marked in the attached file.

Author Response

  1. There are many typos and format errors in the manuscript, some of them have been marked in the attached file.

Response: The following typos are corrected. A comma is added after Thus (line 82), extra spaces removed (111), removed On (line 114), theinhibitory is changed to the inhibitory (line 116), changed compound to compounds (line 133), (b)Substituted is separated with a space (line 143), Compound B is changed to Compound B2 (line 146), C6-C8 is changed to C6-C8 (line 151), mean is changed to the means (line 169), added a period after experiments (line 169), in Table 2 IC50 is changed to IC50 (line 196), changed degree to degrees (line 218), changed reducing of to reducing (line 228), could is changed to can (line 272), physciochemical is changed to physicochemical (line 319), A, B and C compounds were changed to A, B and C (lines 360, 372, and 379), ml is changed to mL (line 362 and 374), 0°C is changed to 0 °C (lines 142, 143, 153, 363, 366, and 381), pyriding is changed to pyridine (line 380), analog is changed analogs (line 444).

  1. Scheme 2, “synthesis of compound B”, however, the scheme is written as synthesis of B2’.

Response: Corrected.

  1. Move Figure 6 to the suitable place, such as, line 282.

Response: Moved.

  1. fluorine is an important element for drug discovery. The promising compound in this work also containing trifluoromethyl group. Thus, recent reviews on fluorine-containing drugs are recommended to be cited.

Response: We agree that trifuoromethyl substituted analogs of niclosamide showed increased potency. However, recent reviews on fluorine-containing drugs are not relevant in our selection of fluorine-containing functional groups in niclosamide analog design and synthesis.

  1. 1H NMR spectra, there are impurity peaks from 1-2 ppm, clean spectra should be provided.

Response: The purity of these compounds was verified by HPLC to be greater than 90% for biological evaluation. The 1H NMR spectra reflect the purity of the compounds in their highest purity obtained in this study.

Reviewer 5 Report

The study provides valuable insights into the potential treatment of enzalutamide and abiraterone resistant prostate cancer using niclosamide analogs. The systematic exploration of the structure-activity relationship of the analogs is a significant step towards the development of a more effective treatment. While the limitations of niclosamide as a treatment for systemic diseases are acknowledged, the identification of analogs with improved properties presents a potential solution. Further research in this area is warranted to determine the clinical utility of these analogs. The work is suitable for publication under medicinal chemistry section of pharmaceuticals journal after making the below mentioned corrections.

1. The introduction of the manuscript is well presented. However it is too lengthy and require mild trimming of this portion by removing the unnecessary content. Additionally, the design part in results and discussion must be shifted to the introduction.

2. There is no result and discussion of the characterization of the compounds. Similarly nothing is presented about the materials and methods of the characterization.

3. 3. D QSAR Study must be   3D-QSAR

Author Response

Reviewer 5

The study provides valuable insights into the potential treatment of enzalutamide and abiraterone resistant prostate cancer using niclosamide analogs. The systematic exploration of the structure-activity relationship of the analogs is a significant step towards the development of a more effective treatment. While the limitations of niclosamide as a treatment for systemic diseases are acknowledged, the identification of analogs with improved properties presents a potential solution. Further research in this area is warranted to determine the clinical utility of these analogs. The work is suitable for publication under medicinal chemistry section of pharmaceuticals journal after making the below mentioned corrections.

  1. The introduction of the manuscript is well presented. However it is too lengthy and require mild trimming of this portion by removing the unnecessary content. Additionally, the design part in results and discussion must be shifted to the introduction.

Response: We have reduced the description of androgen deprivation therapy from the first paragraph of the introduction section. The design description of the chemistry section should remain in the Results and Discussion section.

  1. There is no result and discussion of the characterization of the compounds. Similarly nothing is presented about the materials and methods of the characterization.

Response: Characterization of synthetic molecules includes NMR, MS, HPLC analysis described in Materials and Methods under general information, and in more detail in the supporting information.

  1. 3. D QSAR Study must be 3D-QSAR

Response: Corrected.

Round 2

Reviewer 2 Report

The authors addressed all the comments very carefully. 

Author Response

Thank you.

Reviewer 3 Report

The authors described the design, synthesis, and evaluation of niclosamide analogs as therapeutic agents for enzalutamide-resistant prostate cancer. I consider that the manuscript meets all requirements to be published in “Pharmaceuticals” after minor revision. Although the authors perform most of the modifications, NMR and HRMS spectra of all synthesized compounds should be included in the Supporting Information because it confirms the structure proposed for each compound.

Author Response

The authors described the design, synthesis, and evaluation of niclosamide analogs as therapeutic agents for enzalutamide-resistant prostate cancer. I consider that the manuscript meets all requirements to be published in “Pharmaceuticals” after minor revision. Although the authors perform most of the modifications, NMR and HRMS spectra of all synthesized compounds should be included in the Supporting Information because it confirms the structure proposed for each compound.

Response: We have provided NMR and HRMS spectra of all synthesized compounds in the Supporting information.